# Sleep bolsters schematically incongruent memories

**Jennifer E. Ashton**[1], **Bernhard P. Staresina**[2], **Scott A. Cairney**[1,3]*

**1** Department of Psychology, University of York, York, United Kingdom, **2** Department of Experimental Psychology, University of Oxford, Oxford, United Kingdom, **3** York Biomedical Research Institute, University of York, York, United Kingdom

* scott.cairney@york.ac.uk

**Data Availability Statement:** Study data can be retrieved via the following link: osf.io/vcfgm.

**Funding:** This work was supported by a Medical Research Council (https://mrc.ukri.org/) Career Development Award (MR/P020208/1) to S.A.C. The funders had no role in study design, data

## Abstract

Our ability to recall memories is improved when sleep follows learning, suggesting that sleep facilitates memory consolidation. A number of factors are thought to influence the impact of sleep on newly learned information, such as whether or not we rehearse that information (e.g. via restudy or retrieval practice), or the extent to which the information is consistent with our pre-existing schematic knowledge. In this pre-registered, online study, we examined the effects of both rehearsal and schematic congruency on overnight consolidation. Participants learned noun-colour pairings (e.g. elephant-red) and rated each pairing as plausible or implausible before completing a baseline memory assessment. Afterwards, participants engaged in a period of restudy or retrieval practice for the pairings, and then entered a 12 h retention interval of overnight sleep or daytime wakefulness. Follow-up assessments were completed immediately after sleep or wake, and again 24 h after learning. Our data indicated that overnight consolidation was amplified for restudied relative to retested noun-colour pairings, but only when sleep occurred soon after learning. Furthermore, whereas plausible (i.e. schematically congruent) pairings were generally better remembered than implausible (i.e. schematically incongruent) pairings, the benefits of sleep were stronger for implausible relative to plausible memories. These findings challenge the notion that schema-conformant memories are preferentially strengthened during post-learning sleep.

## Introduction

Memories fade over time, but rehearsing learned materials can help to improve recall. There are two rehearsal strategies that an individual can use to help commit new information to memory: restudy and retrieval practice. For example, one might try to revise for a spelling test by re-reading the words (restudy) or by attempting to recall the words from memory (retrieval practice). In the short term (e.g. within an hour of learning), restudied information is better remembered than information subjected to retrieval practice [1–3]. In the longer-term (e.g. from several hours up to a week after learning), by contrast, the benefits of retrieval practice tend to outweigh those arising from restudy [3,4]. This difference in retention after longer

collection and analysis, decision to publish, or
preparation of the manuscript.

**Competing interests:** The authors have declared
that no competing interests exist.

intervals suggests that restudied memories might be more susceptible to decay than those subjected to retrieval practice.

Memory retention is improved by post-learning sleep [5–11], suggesting that sleep supports the consolidation of newly learned information. Contemporary models of sleep-associated memory processing propose that memories are reactivated during sleep, and thereby integrated into long-term storage [5,12–16]. However, sleep does not benefit all memories equally, with accumulating evidence suggesting that overnight memory gains are more robust for weakly encoded than strongly encoded materials [17–19, but also see 20]. Relatedly, efforts to enhance overnight consolidation via memory cueing in sleep are most effective when pre-sleep learning performance is low [21–23].

Given that sleep may provide the greatest benefit to weakly encoded memories, and that restudied memories are more prone to decay than memories subjected to retrieval practice, restudied information should be particularly responsive to overnight consolidation. Consistent with this view, previous work has indicated that restudied but not retested materials are better remembered after a night of sleep than a day of wakefulness [24,25]. It has thus been suggested that retrieval practice may prompt a rapid consolidation of newly learned information into long-term memory, potentially via similar mechanisms to those underpinning sleep-associated memory processing [26].

Information that is congruent with pre-existing schematic knowledge is typically better remembered than schematically incongruent information [27–29]. The memory benefits of cognitive schemata are thought to arise from interactions between prefrontal cortex and medial temporal lobe, which support efficient learning of schematically congruent materials [30,31]. Interestingly, prior knowledge also enhances sleep-associated consolidation, such that schema-conformant memories are strengthened during sleep to a greater extent than non-conformant memories [32–34]. The interleaved reactivation of new memories and their associated schematic representations during sleep is thought to facilitate the integration of newly learned information into long-term storage [12].

In this pre-registered, online study (osf.io/f82mw), we tested two hypotheses: 1) the benefits of sleep (vs wakefulness) for memory will emerge for restudied memories but not memories subjected to retrieval practice, and 2) the retention advantage for schematically congruent (vs incongruent) memories will be stronger after sleep than wakefulness. We also tested a third novel hypothesis concerning the combined effects of sleep, memory rehearsal and prior knowledge on memory consolidation: assuming that overnight memory processing preferentially strengthens restudied (vs retested) and schematically congruent (vs incongruent) memories, then, after sleep (vs wake), the benefits of restudy (vs retrieval practice) should be greater for schema-conformant than non-conformant information.

We tested our three hypotheses using a source memory paradigm, in which participants learned noun-colour pairings (e.g. elephant-red) and rated the plausibility of each pairing (e.g. a red elephant is implausible). From this plausibility response, we could infer the schematic congruency of each pairing, which was based on each participant's unique understanding of the world. Training (encoding and baseline memory assessment) took place in the morning or evening and was immediately followed by a memory rehearsal phase, during which participants engaged in a period of restudy or retrieval practice for the pairings. Memory for the pairings was re-assessed 12 h later, following a night of sleep (evening training) or a day of wakefulness (morning training). This allowed us to determine the effects of sleep (vs wake) on the consolidation of memories that had been restudied or retested, and were plausible (i.e. schematically congruent) or implausible (i.e. schematically incongruent). Finally, so that we could determine the effects of sleep on memory retention after a longer delay, participants also completed a second follow-up assessment 24 h after training.

## Methods

### Participants

One-hundred and sixty-eight adults were recruited via the online platform Prolific (app.pro-lific.co/). Participants were aged between 18 and 30 years and reported to be living in the UK with English as their first language. On the days of the study, participants were asked to follow their usual daily routines, abstain from alcohol and avoid taking naps (as is standard practice in our lab e.g., [35–38]. Informed consent was obtained from all participants in line with the Research Ethics Committee of the Department of Psychology at the University of York, who approved the study. Participants were required to meet a memory performance criterion in the first session (see below) to continue with the remaining sessions. A total of 102 participants met this criterion and were invited to take part in sessions two and three. Sixty-two participants returned to complete the full study. We analysed data from a final sample of 60 participants after excluding two participants who reported napping during the days of the experiment. The remaining participants took part in a sleep ($n$ = 30, 18 females, mean ± SD age = 25.07 ± 4.02 years) or wake group ($n$ = 30, 25 females, mean ± SD age = 25.00 ± 3.65 years).

Our sample size was calculated using an effect size reported in Antony & Paller (2018) [25]. The effect of interest was an interaction ($\eta_p^2$ = 0.14) from a two-way ANOVA with the factors Delay (AM/PM) and rehearsal strategy (Restudy/Retest). We determined that a minimum sample size of $N$ = 20 would be necessary for 95% power to detect an effect of this magnitude. However, because Antony & Paller (2018) [25] conducted their study in the laboratory, we believed it reasonable to increase our sample to $N$ = 60 to mitigate any noise associated with online data collection. Data collection continued until our desired sample size was met, with each participant providing a full and useable data set.

### Materials and software

Six-hundred and eighty English nouns referring to concrete objects were obtained from the Medical Research Council Psycholinguistics database (websites.psychology.uwa.edu.au/school/mrcdatabase/uwa_mrc.htm). Words were three to eight letters long, with a Kucera–Francis written frequency of 10–100. Only words with concreteness and imageability ratings ranging from 400 to 700 (out of 700) were included [39]. For each participant, 240 nouns were randomly selected for the encoding phase and were each paired with one of four colours (red, yellow, green or blue). An equal number of nouns were paired with each colour (i.e. 60 nouns per colour). The remaining nouns were randomly assigned to each of the three test phases as foils (120 in each test). The experimental tasks were programmed using PsychoPy and hosted on Pavlovia.org [40]. Participants completed the tasks at home on a desktop or laptop (the use of tablets or smartphones was prohibited).

### Procedure overview

The study was divided into three sessions (see Fig 1A). Session one began at 8am (wake group) or 8pm (sleep group), and comprised an encoding phase, a baseline memory test and a memory rehearsal phase. Follow-up memory tests were completed at sessions two and three, which began 12 h and 24 h after session one, respectively. The Stanford Sleepiness Scale [41] and a three-minute psychomotor vigilance task [42,43] were completed at each session to measure participant alertness.

**Encoding.**     There were 240 encoding trials. Each trial began with a fixation cross, presented in the centre of the screen for 500 ms. A randomly selected noun was then displayed

# A. Procedure

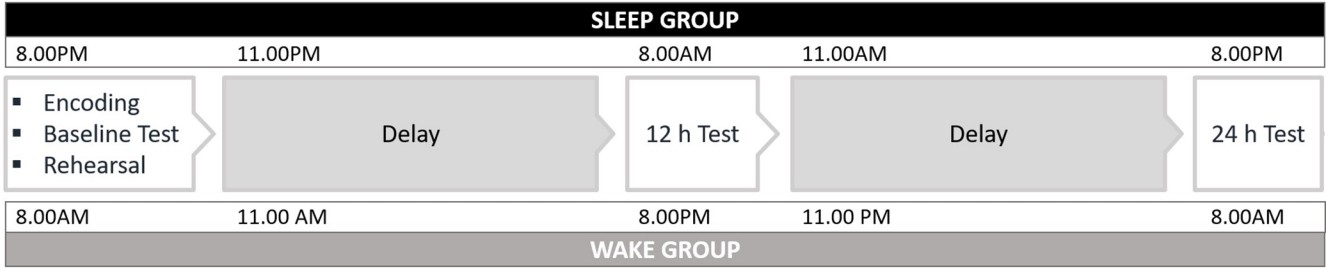

# B. Encoding

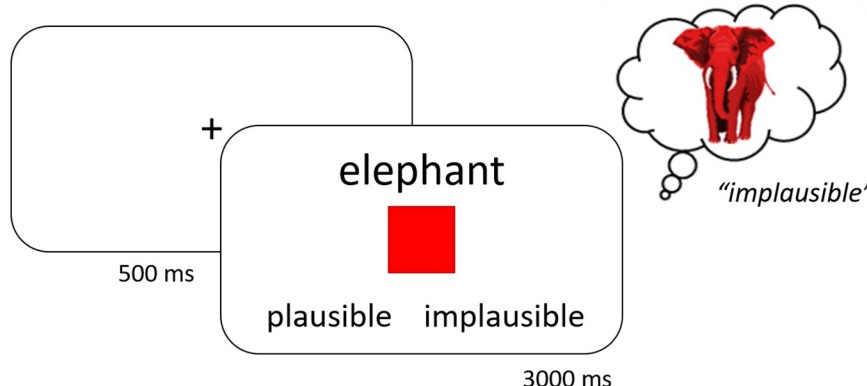

# C. Test

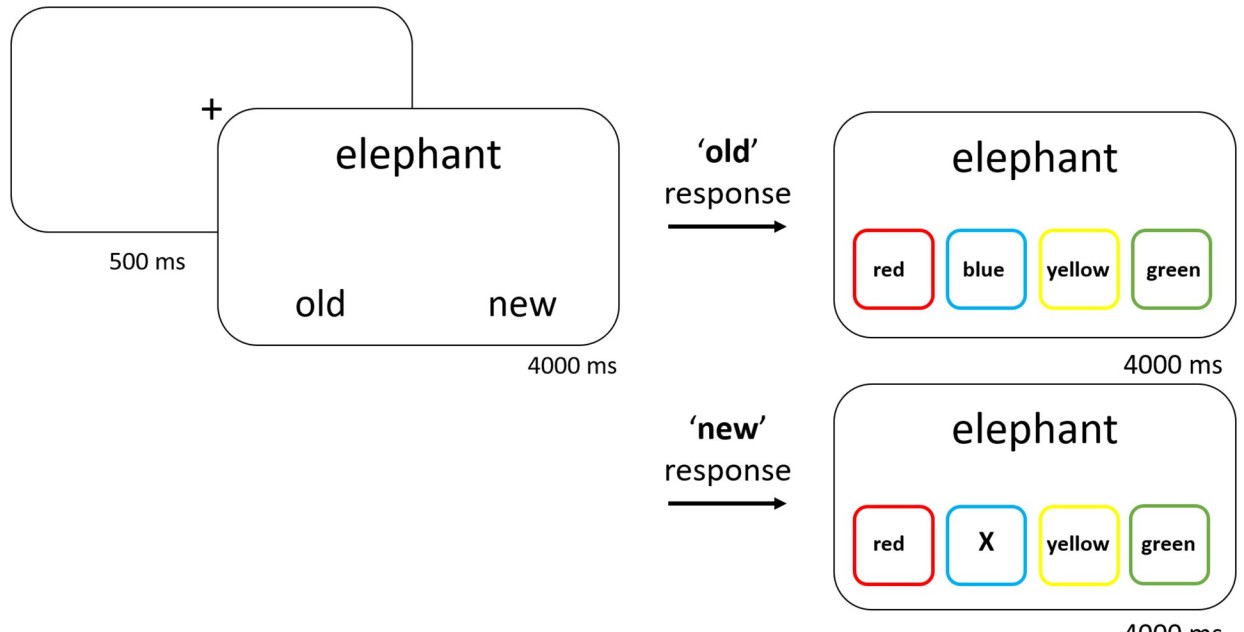

**Fig 1. Experimental procedure and tasks.** (A) Participants completed an encoding phase, a baseline memory test and a memory rehearsal phase in the morning (wake group) or evening (sleep group). Participants returned 12 h and 24 h later to complete follow-up memory tests. (B) On each encoding trial, participants were presented with a noun and a coloured square. Participants were asked to imagine the referent of the noun in the given colour and indicate whether it was a plausible or implausible combination. (C) On each test trial, participants were asked to decide whether a noun was 'old' (i.e. they had seen the

noun at encoding) or 'new' (i.e. they had not seen the noun at encoding). For 'old' responses, participants were asked to select the colour that had appeared with the noun at encoding. For 'new' responses participants were asked to indicate the location of the letter 'X'.

above a coloured square for 3 s (see Fig 1B). Participants were asked to imagine the referent of the noun in the given colour (red, yellow, green or blue) and indicate whether it was a plausible or implausible combination (e.g. a red elephant is implausible). Any trial for which a participant failed to provide a plausible or implausible response was removed from all subsequent analyses. A mean of 3.11% (SEM ± 0.64) trials were removed on this basis. Across both the sleep and wake groups, participants rated 57% (SEM ± 1.41) of the noun-colour pairings as plausible.

**Baseline test.** There were 360 test trials, 240 of which corresponded to the nouns presented at encoding. The remaining 120 trials were unseen foils. On each trial, a central fixation cross (500 ms) preceded a randomly selected noun (see Fig 1C). Participants were asked to indicate whether the noun was 'old' (i.e. they recognised it from encoding) or 'new' (i.e. they did not recognise it from encoding). For each 'old' response, participants were also asked to select the colour that had been paired with the noun at encoding, and guess if they were not certain. For each 'new' response, participants were asked to indicate the location of a letter 'X' that was randomly presented in one of four locations. Employing this two-step approach provided an index of item memory (noun recognition) and source memory (colour recall). Note that we refer to colour recall as source memory because the colour associated with each noun reflects the context or source with which that noun was encountered at learning.

To motivate engagement with the task, participants were provided with an overall performance score at the end of the baseline test (calculated as the percentage of correct old/new responses) and were encouraged to try and improve on their score in the next test.

**Performance criteria.** If a participant did not meet the following baseline performance criteria, they were not invited back for the follow-up tests and a new participant was recruited in their place. We excluded 66 participants on this basis.

The performance criteria were set in two steps:

1. Item memory score of 50% or above. Calculated as the proportion of correctly recognised 'old' nouns, minus the proportion of foils incorrectly rated as 'old'.

2. Source memory score of 40% or above. Calculated as the proportion of correctly identified colours (restricted to correctly recognised nouns), referred to hereafter as colour hits.

**Memory rehearsal.** After the baseline test, participants completed two rehearsal phases: restudy and retrieval practice (order counterbalanced across participants). Noun-colour pairings that were correctly retrieved at baseline were randomly allocated to three conditions: restudy, retrieval practice and a no-rehearsal control condition. The restudy and retrieval practice phases followed the same procedures as at encoding and the baseline test, respectively. Participants received only one additional exposure to each noun-colour pairing allocated to the restudy and retrieval practice conditions. Pairings in the no-rehearsal condition were not revisited until the follow-up tests.

To check whether there was any bias in the distribution of noun-colour pairings between conditions, we applied colour hits at the baseline test to a 2 (Plausibility: Plausible/Implausible) * 3 (Rehearsal Strategy: Restudy/Retrieval Practice/No-Rehearsal) * 2 (Group: Sleep/Wake) mixed ANOVA. There was a main effect of Plausibility ($F(1, 58) = 21.73$, $p < .001$, $\eta_p^2 = 0.27$), reflecting participants' tendency to rate more noun-colour pairings as plausible than

**Table 1. Source memory scores (colour hits) at the baseline test for the sleep and wake groups.** Scores are presented separately for each plausibility condition. Data are presented as mean ± SEM.

|  | Plausible | Implausible |
|---|---|---|
| **Sleep** | 72.99 ± 2.26 | 60.85 ± 2.13 |
| **Wake** | 71.79 ± 2.23 | 59.10 ± 2.55 |

implausible at encoding. There was also a main effect of Rehearsal Strategy ($F(2,116) = 3.13$, $p = .048$, $\eta_p^2 = 0.05$), reflecting a slight difference in the number of pairings allocated to each rehearsal condition (mean ± SEM, Restudy: 21.08 ± 0.64, Retrieval Practice: 20.82 ± 0.65, No-Rehearsal: 20.83 ± 0.66). There was no main effect of Group and none of the interactions were significant ($p > .05$).

The main effect of Rehearsal Strategy emerged because of a randomisation error, where on occasions that the number of baseline colour hits was not divisible by 3, the first 'additional' noun-colour pairing was always allocated to the Restudy condition. For example, if 61 noun-colour pairings were correctly recalled at baseline, then 20 would be allocated to Retrieval Practice, 20 would be allocated to No-Rehearsal and the remaining 21 would be allocated to Restudy.

**Follow-up tests.** Participants completed a follow-up test 12 h after the baseline test (i.e. after a night of sleep [sleep group] or a day of wakefulness [wake group]). A second follow-up test took place 24 h after the baseline assessment. Both follow-up tests followed the same procedures as the baseline test, with the exception that a new set of foils were used in each. Participants were not asked if they had undertaken any active rehearsal of the noun-colour pairings during the intervals between experimental sessions.

## Statistical analysis

**Item memory.** D-prime [normalised (hits / hits + misses)–normalised (false alarms / false alarms + correct rejections)] was calculated to assess recognition accuracy for nouns (item memory) at the baseline test and the 12/24 h follow-ups. Independent samples t-tests comparing d-prime between the sleep and wake groups were performed at each test.

**Source memory.** Our main dependent variable of interest was source memory retention, calculated separately for the 12 h and 24 h follow-ups as the number of colour hits divided by the number of colour hits at baseline (converted to percentages). Data at each follow-up was applied to a 2 (Group: Sleep/Wake) * 3 (Rehearsal Strategy: Restudy/Retrieval Practice/No-Rehearsal) * 2 (Plausibility: Plausible/Implausible) mixed ANOVA. The plausibility of each noun-colour pairing was determined by the plausible/implausible responses provided at encoding. Significant interactions were interrogated using post-hoc comparisons with a Bonferroni-corrected alpha adjustment for the number of tests performed. All statistical analyses were performed in R [44].

## Results

### Item memory

There was no difference in item memory (noun recognition accuracy, d-prime) between the sleep and wake groups at the baseline test (sleep group mean ± SEM: 2.09 ± 0.07, wake group: 2.11 ± 0.10, $t(58) = 0.21$, $p = .839$). However, at the 12 h follow-up, item memory was higher for participants who had slept (2.00 ± 0.10) relative to those who had remained awake (1.71 ± 0.11, $t(58) = 2.01$, $p = .049$, $d = 0.52$). No between-group difference was present at the 24 h follow-up (sleep group: 2.01 ± 0.10, wake group: 1.97 ± 0.13, $t(58) = 0.26$, $p = 0.80$).

**Table 2. Source memory scores (%) for the 12 h and 24 h follow-ups in the sleep and wake groups.** Scores are presented separately for each rehearsal strategy and plausibility condition. Data are presented as mean ± SEM.

| | Restudy | | Retrieval Practice | | No-Rehearsal | |
|---|---|---|---|---|---|---|
| | **Plausible** | **Implausible** | **Plausible** | **Implausible** | **Plausible** | **Implausible** |
| *12 h Follow-up* | | | | | | |
| Sleep | 86.99 ± 2.01 | 81.35 ± 3.12 | 82.26 ± 2.11 | 73.05 ± 3.15 | 73.28 ± 2.48 | 60.55 ± 2.46 |
| Wake | 77.49 ± 3.04 | 66.40 ± 4.34 | 75.11 ± 2.51 | 64.05 ± 3.52 | 62.93 ± 2.78 | 44.29 ± 3.38 |
| *24 h Follow-up* | | | | | | |
| Sleep | 80.25 ± 3.00 | 74.56 ± 3.72 | 76.37 ± 3.01 | 64.66 ± 3.71 | 67.37 ± 3.67 | 54.38 ± 3.32 |
| Wake | 79.41 ± 2.83 | 67.80 ± 3.78 | 76.74 ± 2.60 | 68.07 ± 2.67 | 63.46 ± 2.84 | 47.84 ± 3.43 |

## Source memory: Baseline

Source memory scores (colour hits) at baseline (see Table 1) were applied to a 2 (Group: Sleep/Wake) $*$ 2 (Plausibility: Plausible/Implausible) mixed ANOVA. There was a main effect of Plausibility ($F(1, 58) = 127.22$, $p < .001$, $\eta_p^2 = 0.69$), as plausible noun-colour pairings were better remembered than implausible pairings. However, there was no main effect of Group ($F(1, 58) = 0.23$, $p = .630$) and no interaction between factors ($F(1, 58) = 0.06$, $p = .802$).

## Source memory: Sleep and rehearsal

We first tested the hypothesis that a memory benefit of sleep would emerge for restudied source memories but not source memories subjected to retrieval practice.

**12 h follow-up.** There was a main effect of Group ($F(1, 58) = 11.39$, $p = .001$, $\eta_p^2 = 0.16$), indicating that source memory retention was significantly higher after a night of sleep than a day of wakefulness (see Table 2 and Fig 2). There was also a main effect of Rehearsal Strategy ($F(2, 116) = 76.53$, $p < .001$, $\eta_p^2 = 0.57$): retention scores were higher in 1) the restudy vs no-rehearsal condition ($t(59) = 12.58$, $p < .001$, $d = 1.62$), 2) the retrieval practice vs no-rehearsal condition ($t(59) = 8.51$, $p < .001$, $d = 1.10$), and 3) the restudy vs retrieval practice condition ($t(58) = 2.90$, $p = .005$, $d = 0.38$), indicating that restudy produced the greatest overall retention benefit. However, there was no Group $*$ Rehearsal Strategy interaction ($F(2, 116) = 1.70$, $p = .187$), indicating that the memory effects of sleep (vs wake) were not amplified in any of the rehearsal conditions.

**24 h follow-up.** There was no main effect of Group at the 24 h follow-up ($F(1, 58) = 0.39$, $p = .535$), but the main effect of Rehearsal Strategy remained ($F(2, 116) = 83.31$, $p < .001$, $\eta_p^2 = 0.59$): retention scores were again higher in 1) the restudy vs no-rehearsal condition ($t(58) = 13.72$, $p < .001$, $d = 1.77$), 2) the retrieval practice vs no-rehearsal condition ($t(58) = 8.81$, $p < .001$, $d = 1.14$), and 3) the restudy vs retrieval practice condition ($t(58) = 2.67$, $p = .010$, $d = 0.35$). Interestingly, a significant Group $*$ Rehearsal Strategy interaction also emerged at the 24 h follow-up ($F(2, 116) = 3.63$, $p = .029$, $\eta_p^2 = 0.06$, see Fig 3). Whereas source memory retention was significantly higher after restudy than retrieval practice in the sleep group ($t(29) = 4.11$, $p < .001$, $d = .75$), no such difference emerged in the wake group ($t(29) = 0.49$, $p = .626$). Hence, after a 24 h delay, the retention advantage for restudied (vs retested) information was only evident among individuals who had slept soon after learning.

**Subsidiary analyses.** Although the foregoing analyses of source memory retention were restricted to colours for which the associated noun was correctly recognised at baseline, they did not control for noun recognition (item memory) at the 12 h or 24 h follow-ups. This is an important consideration, because if a participant failed to re-recognise a noun at the 12 h follow-up, the trial was scored as a source memory error, meaning that any effect of sleep on item

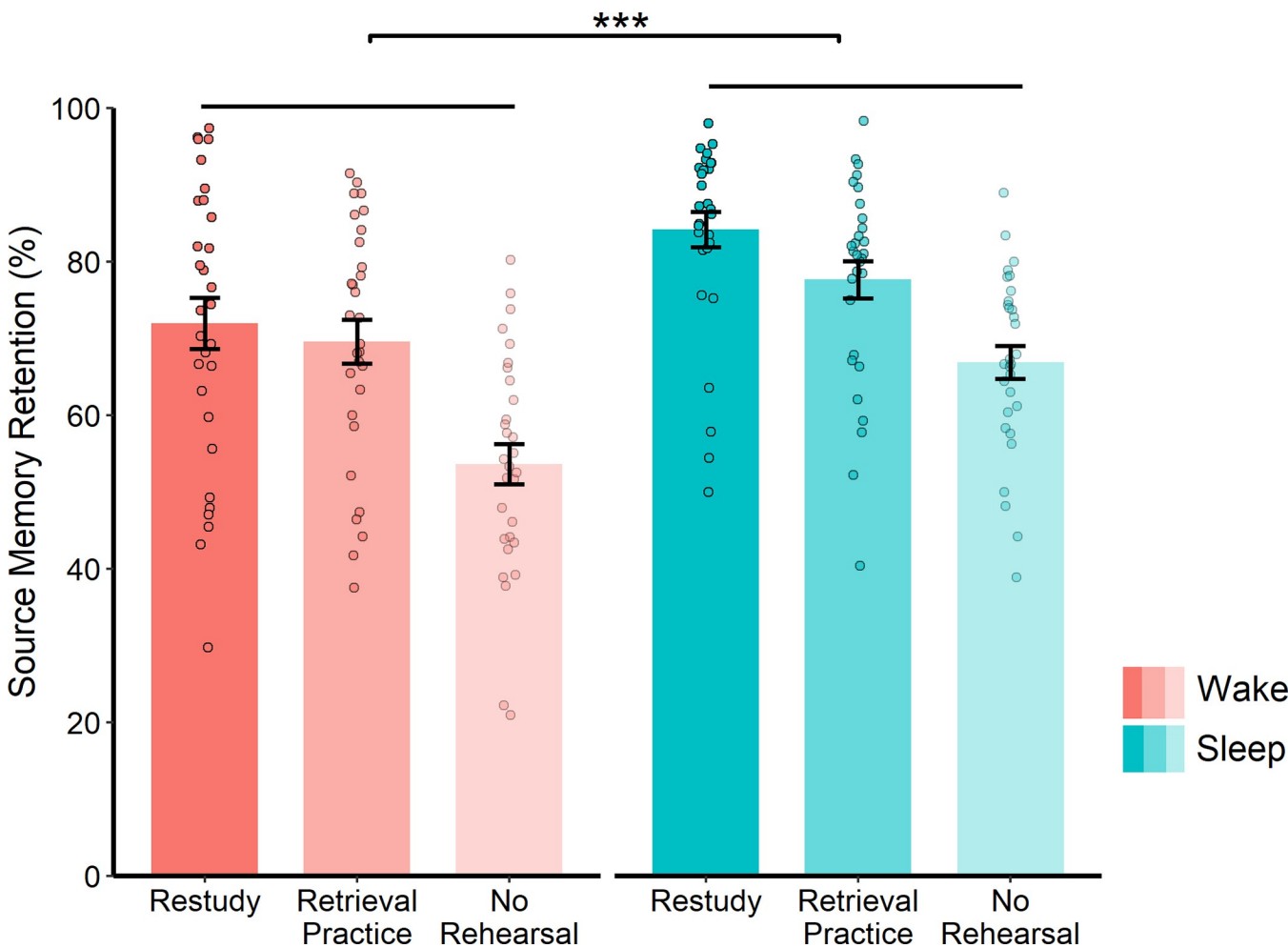

**Fig 2. Source memory retention and rehearsal, 12h follow-up.** Source memory retention at the 12 h follow-up for the sleep and wake groups. Data is shown for each rehearsal condition, collapsed across the plausibility conditions. Error bars represent SEM. *** $p < .001$.

memory could have been misinterpreted as an effect of sleep on source memory. To address this concern, we repeated the above analyses (and those described in the next section), but fully controlled for item memory by restricting the input data to colours for which the associated noun was correctly recognised at both the baseline test and the relevant 12 h or 24 h follow-up (see S1 File). In brief, these subsidiary analyses confirmed the results of our main analyses, although the Group * Rehearsal Strategy interaction at the 24 h follow-up became non-significant ($F(2,116) = 2.47$, $p = .089$, $\eta_p^2 = 0.04$).

### Source memory: Sleep and schematic congruency

Next, we tested the hypothesis that overnight consolidation strengthens schematically congruent (plausible) source memories to a greater extent than schematically incongruent (implausible) source memories.

**12 h follow-up.** There was a main effect of Plausibility ($F(1, 58) = 109.64$, $p < .001$, $\eta_p^2 = 0.65$), indicating that source memory retention was higher for plausible relative to implausible noun-colour pairings (see Fig 4A). Importantly, there was also a significant Group * Plausibility interaction ($F(1, 58) = 4.10$, $p = .048$, $\eta_p^2 = 0.07$), but not in the direction predicted. Post-

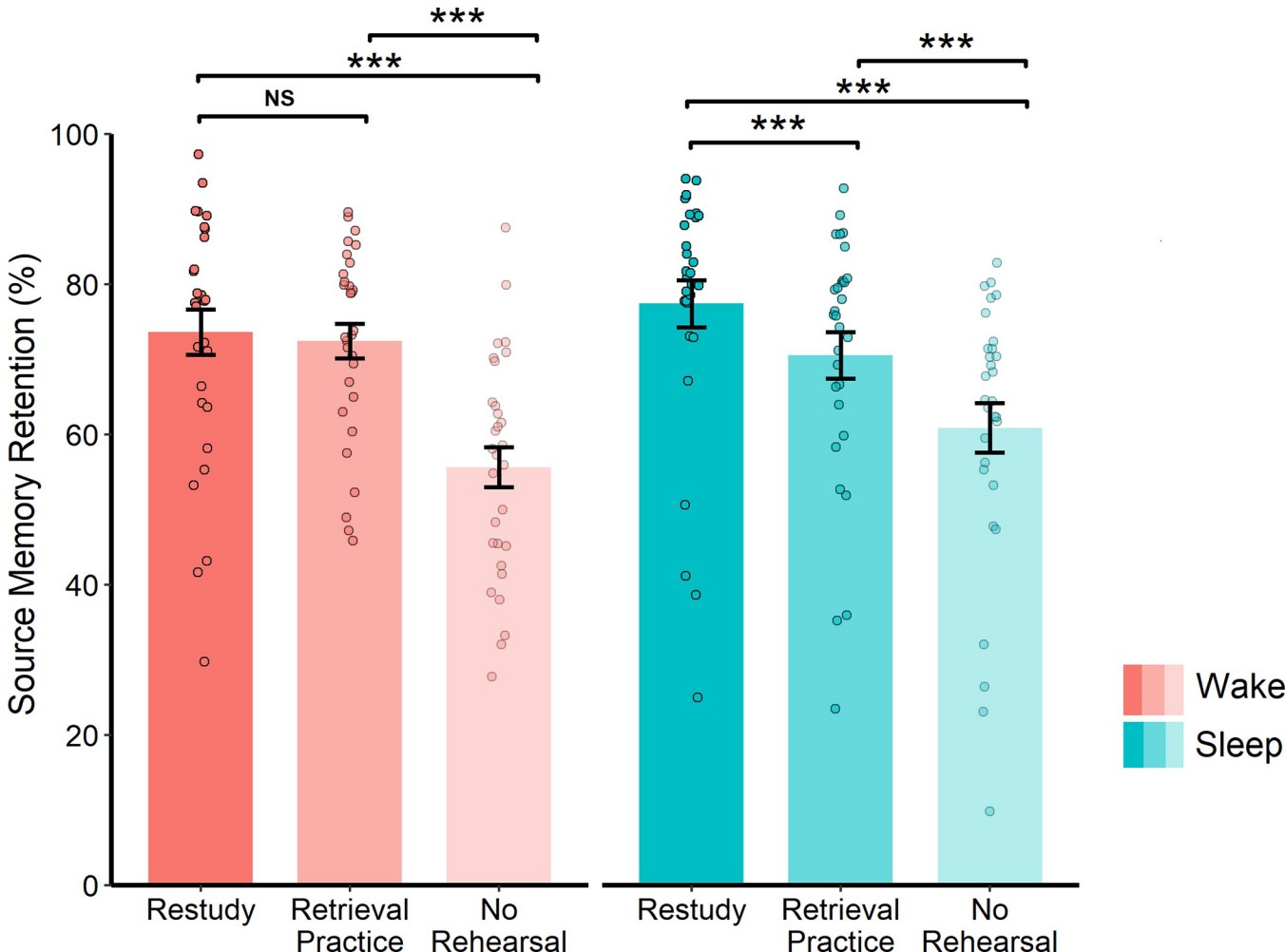

**Fig 3. Source memory retention and rehearsal, 24-hour follow-up.** Source memory retention at the 24 h follow-up for the sleep and wake groups. Data is shown for each rehearsal condition, collapsed across the plausibility conditions. Error bars represent SEM. *** $p \leq$ .001, NS = not significant.

hoc comparisons revealed that the retention advantage for plausible (vs implausible) pairings was *smaller* in the sleep group ($t(29)$ = 6.18, $p <$ .001, $d$ = 1.13) than the wake group ($t(29)$ = 8.56, $p <$ .001, $d$ = 1.56, see Fig 4A). Moreover, the memory benefits of sleep (vs wakefulness) were stronger for implausible pairings ($t(58)$ = 3.41, $p$ = .001, $d$ = .88) than plausible pairings ($t(58)$ = 3.00, $p$ = .004, $d$ = .78). Taken together, these findings suggest that sleep preferentially facilitates the consolidation of implausible (and schematically incongruent) associations.

**24 h follow-up.** The main effect of Plausibility was still present at the 24 h follow-up ($F(1, 58)$ = 87.36, $p <$ .001, $\eta_p^2$ = 0.60), but the Group * Plausibility interaction no longer remained ($F(1, 58)$ = 0.61, $p$ = .439, see Fig 4B). However, under the assumption that sleep preferentially strengthens implausible associations, this null effect is to be expected, as participants in the wake group had also slept before the 24 h follow-up. To test this possibility, we carried out an exploratory analysis of source memory retention between the 12 h and 24 h follow-ups. If sleep facilitates the consolidation of schematically incongruent associations, then the retention of implausible (vs plausible) pairings should be better in the wake group than the sleep group, because only the wake group have slept between these two time points.

## A. 12 h Follow-up

## B. 24 h Follow-up

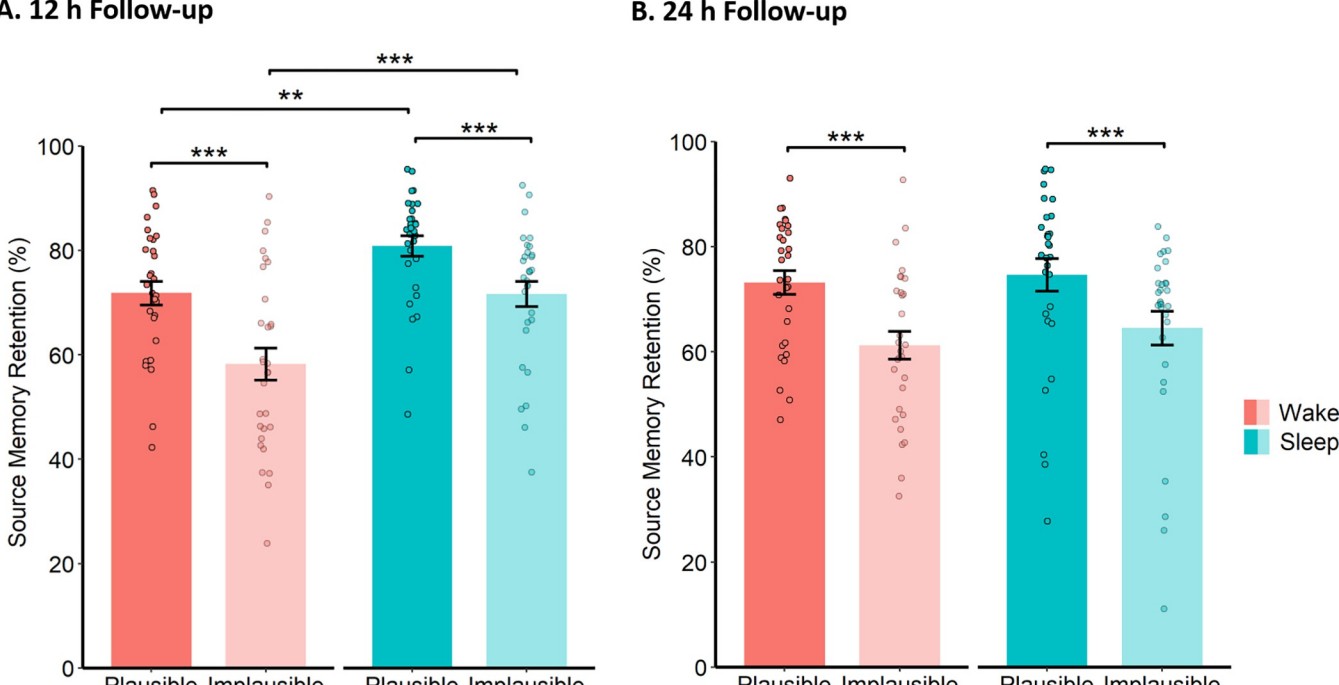

**Fig 4. Source memory retention and schematic congruency.** Memory retention at the (A) 12 h and (B) 24 h follow-ups for the plausible and implausible conditions. Data are collapsed across the rehearsal conditions. Error bars represent SEM. *** $p \leq .001$, ** $p \leq .01$.

Source memory retention scores were recalculated as the number of colour hits at the 24 h follow-up divided by the number of colour hits at the 12 h follow-up (converted to percentages, hence a score > 100% indicates an improvement between the two tests) and then reapplied to the ANOVA described above. Consistent with the memory benefits of sleep observed at the 12 h follow-up, a main effect of Group emerged in this new ANOVA, with the wake group outperforming the sleep group ($F(1, 58) = 19.47$, $p < .001$, $\eta_p^2 = 0.25$).

Importantly, a significant Group * Plausibility interaction was also observed ($F1, 58) = 8.23$, $p = .006$, $\eta_p^2 = 0.12$). Whereas retention was higher for implausible (vs plausible) associations in the wake group ($t(29) = 2.84$, $p = .008$, $d = .52$), no such difference emerged in the sleep group ($t(29) = 0.92$, $p = .366$, see Fig 5). Taken together with the results of our main analysis, these findings suggest that overnight consolidation preferentially strengthens schematically incongruent memories. It is worth noting that the wake group even showed gains in retrieval performance between the 12 h and 24 h follow-ups (i.e. retention scores > 100%), which were especially pronounced for implausible noun-colour pairings, whereas the sleep group showed forgetting across both plausible and implausible pairings (thus explaining the non-significant Group main effect in our main analysis of the 24 h follow-up). There was no main effect of Rehearsal Strategy ($F(2,116) = 0.14$, $p = 0.87$), no Group * Rehearsal Strategy interaction ($F(2,116) = 0.69$, $p = .505$) and no Rehearsal Strategy * Plausibility interaction ($F(2,116) = 1.16$, $p = .318$) in this exploratory analysis.

### Sleep, rehearsal and schematic congruency

Finally, we tested the hypothesis that sleep-associated consolidation is enhanced for schematically congruent memories that have been restudied prior to sleep. However, there was no significant Group * Rehearsal Strategy * Plausibility interaction at either the 12 h ($F(2,116) =$

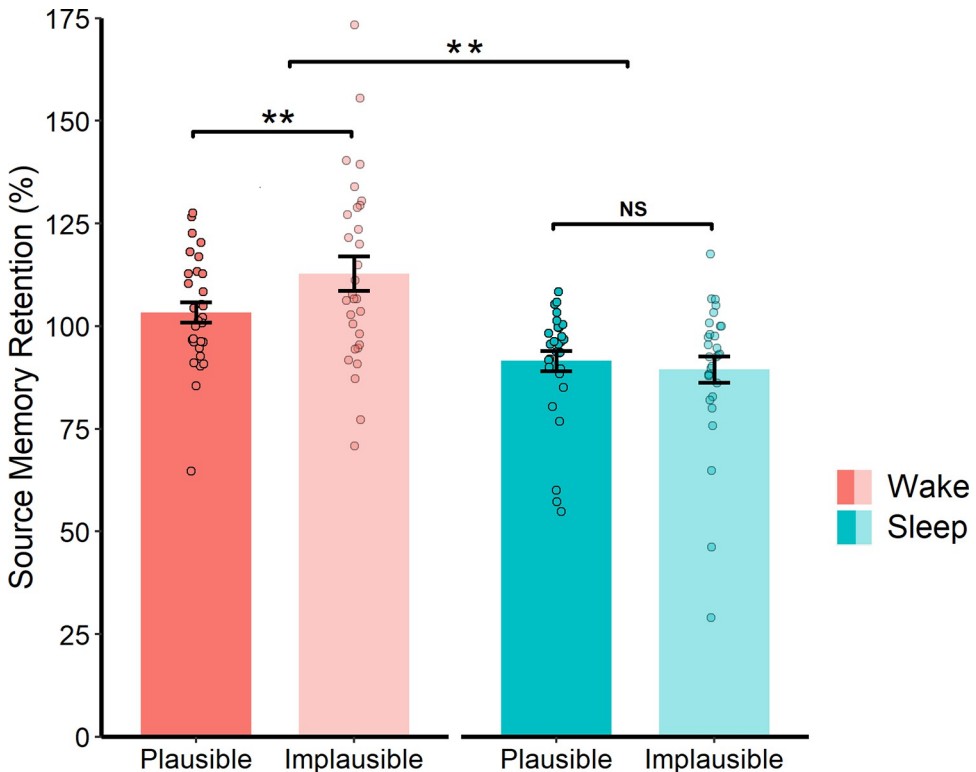

**Fig 5. Source memory retention and schematic congruency (recalculated at 24 h).** Memory retention at the 24 h follow-up with retention scores recalculated as the proportion of retention at the 12 h follow-up (scores > 100% indicate an improvement between the time points). Data are collapsed across the rehearsal conditions. Error bars represent SEM. ** $p \leq .01$, NS = not significant.

0.31, $p = .736$) or 24 h follow-up ($F(2,116) = 1.37$, $p = .258$). A significant Rehearsal Strategy * Plausibility interaction was observed at the 12 h follow-up $F(2, 116) = 0.31$, $p = .029$, $\eta_p^2 = 0.06$): the retention advantage for plausible (vs implausible) pairings was stronger in the no-rehearsal condition ($t(59) = 7.60$, $p < .001$, $d = 0.98$) than the retrieval practice ($t(59) = 6.30$, $p < .001$, $d = 0.81$) or restudy conditions ($t(59) = 3.83$, $p < .001$, $d = 0.50$). This interaction was not present at the 24 h follow-up ($p > .05$).

## Self-reported sleep

Participants reported the number of hours that they had slept between rehearsal and the 12 h follow-up (sleep group) or between the 12 h follow-up and the 24 h follow-up (wake group). Hours slept in the sleep group (mean ± SEM, 6.70 ± 0.26) did not significantly correlate with source memory retention scores in any of the Rehearsal Strategy/Plausibility conditions at the 12 h follow-up ($p > .05$). There were also no significant relationships between hours slept in the wake group (7.37 ± 0.18) and source memory retention scores at the 24 h follow-up ($p > .05$).

## Alertness and vigilance

**Stanford sleepiness scale.** Scores on the Stanford Sleepiness Scale (see Table 3) were subjected to a 2 (Group: Sleep/Wake) * 3 (Session: Encoding/12 h/24 h) mixed ANOVA. There was a main effect of Session ($F(2, 116) = 5.04$, $p = .008$, $\eta_p^2 = 0.08$), with participants rating themselves as more alert at encoding than at the 12 h follow-up ($t(59) = 3.07$, $p = .003$,

**Table 3. Scores on the Stanford Sleepiness Scale (SSS) and the proportion of attentional lapses in the psychomotor vigilance task (PVT) for each session. Data are presented as mean (± SEM).**

| | Encoding | | 12 h Follow-up | | 24 h Follow-up | |
|---|---|---|---|---|---|---|
| | SSS | PVT Lapse | SSS | PVT Lapse | SSS | PVT Lapse |
| **Sleep** | 3.07 ± 0.19 | 15.98 ± 5.07 | 3.67 ± 0.24 | 11.94 ± 3.48 | 2.97 ± 0.27 | 9.77 ± 2.96 |
| **Wake** | 2.57 ± 0.19 | 5.31 ± 1.11 | 3.10 ± 0.24 | 5.53 ± 1.38 | 2.90 ± 0.21 | 9.49 ± 2.93 |

$d = 0.40$). All other between-session differences were non-significant ($p > .05$). There was no main effect of Group ($F(1, 58) = 2.65$, $p = .109$) and no Group * Session interaction ($F(2, 116) = 1.04$, $p = .358$).

**Psychomotor vigilance task.** Attentional lapses (reaction times > 500 ms [45]) on the psychomotor vigilance task (see Table 3) were subjected to a 2 (Group: Sleep/Wake) * 3 (Session: Encoding/12 h/24 h) mixed ANOVA. One participant was excluded from this analysis as no responses were recorded for one of their sessions. There were no main effects of Group ($F(1, 57) = 2.24$, $p = .140$) or Session ($F(2, 144) = 0.50$, $p = .606$). A Group * Session interaction was observed ($F(2, 114) = 3.75$, $p = .027$, $\eta_p^2 = .06$), with participants in the sleep group having more attentional lapses at encoding than participants in the wake group ($t(57) = 2.02$, $p = .048$, $d = .53$). However, this between-group difference did not survive a Bonferroni-corrected alpha threshold of $p \leq .006$.

## Discussion

We assessed the impact of rehearsal and prior knowledge on sleep-associated memory consolidation in a pre-registered, online study. In keeping with the previously reported benefits of sleep for offline memory processing, source memory retention was higher at the 12 h follow-up in the sleep group than the wake group (an advantage that had subsided by the 24 h follow-up once participants in the wake group had slept). Although rehearsal strategy had no effect on the memory benefits of sleep at the 12 h follow-up, a significant interaction emerged at the 24 h follow-up, such that restudied memories were better remembered than retested memories, but only in the sleep group (i.e. those individuals who had slept soon after learning). However, this interaction became non-significant in a subsidiary analysis that controlled for item memory performance at the baseline test and 24 h follow-up. Prior knowledge also influenced overnight memory processing, but in the opposite direction to that predicted by our hypothesis. The benefits of sleep at the 12 h follow-up were stronger for implausible (i.e. schematically incongruent) than plausible (i.e. schematically congruent) memories. This selective influence of sleep on implausible memories was further demonstrated in an exploratory analysis of the 24 h follow-up, where the wake group (having now slept) showed a memory advantage for implausible relative to plausible information. Our findings therefore demonstrate that overnight consolidation preferentially strengthens memories that do not conform to pre-existing, schematic knowledge.

That only the sleep group exhibited greater retrieval of restudied relative to retested memories at the 24 h follow-up is in keeping with earlier studies assessing the impact of rehearsal strategy on sleep-associated consolidation [24,25]. It has been suggested that retrieval practice provides a fast route to consolidation by rapidly strengthening newly acquired memories, potentially via similar mechanisms to those underpinning the memory benefits of sleep [26]. Overnight memory processes may therefore bypass previously retrieved memories, prioritising the stabilisation of restudied, and proportionately unconsolidated, information.

Previous studies have demonstrated that overnight consolidation effects are strongest for information learned within a few hours of sleep [10,46–48]. Our findings build on this prior

work by suggesting that the selective benefits of sleep for restudied information are contingent on sleep taking place soon after learning. Future online studies can assess this possibility more directly by asking participants to report their bed times and determining whether sleep-associated memory gains for restudied information are amplified among individuals who go to bed after a short (vs long) post-rehearsal delay. This could have potentially important practical implications for learning and education, such as the optimal timing of study sessions when students are preparing for examinations.

If the benefits of sleep for restudied information are contingent on sleep taking place soon after learning, then one might have expected to observe a selective effect of sleep on restudied noun-colour pairings at both the 12 h and 24 h follow-ups. It is worth noting, however, that although a significant Group * Rehearsal Strategy interaction did not emerge at the 12 h follow-up, the pattern of results was very similar to that of the 24 h follow-up (where a significant Group * Rehearsal Strategy interaction did occur), with sleep providing the strongest benefit for restudied noun-colour pairings (see Figs 2 and 3). Interestingly, our exploratory analysis of the 24 h follow-up (with retention calculated relative to the 12 h follow-up) showed a main effect of Group, indicating that the wake group (who had slept between the 12 h and 24 h follow-ups) outperformed the sleep group (who had remained awake across the same interval). Yet, this analysis did not reveal a significant Group * Rehearsal Strategy interaction, suggesting that the memory benefits of delayed sleep are unaffected by prior re-engagement with the learned materials.

A further point to consider is that participants' memory for all noun-colour pairings was assessed at both the 12 h and 24 h follow-ups, which meant that pairings in the restudy condition had undergone both restudy and retrieval practice by the time of the 24 h assessment. We therefore cannot rule out the possibility that the selective influence of sleep on restudied information at the 24 h follow-up was influenced by retrieval at the 12 h follow-up. We chose to test all noun-colour pairings at both follow-ups to ensure that there would be a sufficient number of stimuli for the various rehearsal conditions. For example, despite participants encoding 240 noun-colour pairings, only 125.43 (mean ± SEM, ± 3.87) of these were correctly retrieved at baseline, meaning that each rehearsal condition contained 41.81 (± 0.29) pairings. As an alternative, we could have assessed retention for half of the noun-colour pairings at the 12 h follow-up and the other half at the 24 h follow-up, but this would have left only ~21 pairings per rehearsal condition, which would have been further subdivided by the plausibility condition. We therefore believe that our chosen approach was optimal, given the various research questions that we sought to address. Future work focusing on the relationship between overnight consolidation and rehearsal (independent of schematic congruency) can build on our preliminary findings by assessing the isolated effects of restudy and retrieval practice on sleep-associated memory processing.

Our observation that memory retention was improved for plausible (i.e. schematically congruent) relative to implausible (i.e. schematically incongruent) noun-colour pairings is in keeping with a large body of work on the memory benefits of cognitive schemata [49,50]. However, in direct contrast to our hypothesis, we found that sleep-associated memory gains were amplified for implausible relative to plausible pairings. We observed this effect in both the sleep and wake groups after isolating memory retention across their respective intervals of overnight sleep, suggesting that sleep may offer the greatest advantage to memories that deviate from existing knowledge.

The preferential impact of sleep on schematically incongruent memories appears to oppose theoretical frameworks arguing that sleep actively underpins both schema formation and the addition of new knowledge into existing schemata [12]. Because schematically incongruent declarative associations are thought to be stored as episodic representations in medial temporal

lobe [31], and newly formed episodic memories are thought to be reactivated in hippocampus during post-learning sleep [5], it is possible that schematically incongruent episodic memories are more receptive to overnight consolidation, as compared to schematically congruent information. Along these lines, it is worth noting that previous studies reporting enhanced overnight memory gains for schema-conformant information have used very different memory paradigms to the current study, which include melodic [32] or spatial navigation tasks [33].

Memory retention at baseline was generally poorer for implausible than plausible noun-colour pairings, suggesting that the encoding of implausible pairings was relatively weak. Because previous work has shown that the memory benefits of sleep are more robust for weakly relative to strongly encoded materials [17–19, but also see 20], amplified overnight memory gains for implausible (vs plausible) pairings might reflect an impact of encoding strength, rather than schematic congruency, on sleep-associated consolidation. Yet, if it were the case that implausible pairings were selectively strengthened during sleep because they were weakly encoded, then a three-way interaction should have emerged in our Group * Rehearsal Strategy * Plausibility ANOVA, with the weakest memories (i.e. non-rehearsed and implausible noun-colour pairings) showing the greatest enhancement across overnight sleep, but this was not the case.

In conclusion, our online study adds to an extensive body of laboratory-based literature demonstrating that sleep supports offline memory processing [5–8].Furthermore, our data suggest that: 1) the memory benefits of sleep are enhanced for restudied information (as compared to information that has undergone retrieval practice), but only when sleep occurs soon after learning, and 2) sleep-associated memory gains are bolstered for schematically incongruent (relative to schematically congruent) associations. These findings provide new insights into the interaction of prior knowledge, online rehearsal strategies and offline consolidation processes.

## Supporting information

**S1 File. Supplementary results.**
(DOCX)

## Acknowledgments

Study data can be retrieved via the following link: http://osf.io/vcfgm.

## Author Contributions

**Conceptualization:** Bernhard P. Staresina.

**Data curation:** Jennifer E. Ashton.

**Formal analysis:** Jennifer E. Ashton.

**Funding acquisition:** Scott A. Cairney.

**Investigation:** Jennifer E. Ashton.

**Methodology:** Jennifer E. Ashton, Bernhard P. Staresina, Scott A. Cairney.

**Project administration:** Scott A. Cairney.

**Resources:** Scott A. Cairney.

**Writing – original draft:** Jennifer E. Ashton.

**Writing – review & editing:** Bernhard P. Staresina, Scott A. Cairney.

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
