## [Decision Letter · Decision Letter 0]

7 Dec 2021

PONE-D-21-35298Sleep bolsters schematically incongruent memoriesPLOS ONE

Dear Dr. Cairney,

Thank you for submitting your manuscript to PLOS ONE. After careful consideration, we feel that it has merit but does not fully meet PLOS ONE’s publication criteria as it currently stands. Therefore, we invite you to submit a revised version of the manuscript that addresses the points raised during the review process.

We look forward to receiving your revised manuscript.

Kind regards,

Maria Wimber

Academic Editor

PLOS ONE

Journal Requirements:

"This work was supported by a Medical Research Council Career Development Award

(MR/P020208/1) to S.A.C. Study data can be retrieved via the following link: osf.io/vcfgm"

"This work was supported by a Medical Research Council (https://mrc.ukri.org/) Career Development Award (MR/P020208/1) to S.A.C. The funders had no role in study design, data collection and analysis, decision to publish, or preparation of the manuscript."

Additional Editor Comments:

As can be seen from the reviewers’ comments, both experts find merit in this study but have major concerns regarding the analyses and interpretation of the results. These points need to be addressed in a major revision of the manuscript, including additional analyses.

Reviewers' comments:

Reviewer's Responses to Questions

**Comments to the Author**

1. Is the manuscript technically sound, and do the data support the conclusions?

Reviewer #1: Partly

Reviewer #2: Yes

2. Has the statistical analysis been performed appropriately and rigorously? 

Reviewer #1: Yes

Reviewer #2: Yes

3. Have the authors made all data underlying the findings in their manuscript fully available?

Reviewer #1: Yes

Reviewer #2: Yes

4. Is the manuscript presented in an intelligible fashion and written in standard English?

Reviewer #1: Yes

Reviewer #2: Yes

5. Review Comments to the Author

Reviewer #1: In this manuscript, Ashton and colleagues investigate the role of rehearsal strategy, schema congruency and sleep on cued recall of colors previously associated with nouns. In an online study, two groups of participants learned noun-color pairs by deciding whether the pairing was plausible or not either in the morning or in the evening. Their memory was tested at 12h and 24h delays, showing the expected main effects at the 12h interval. At 24h, the difference between restudy and retrieval remained only for the sleep group, which according to the authors indicates a critical time window shortly after learning for sleep-related benefits on restudied information. Furthermore, a significant interaction revealed group differences between memory for plausible and implausible items. The manuscript is concise and well written, however I see several issues, especially concerning interpretation of the results.

24h sleep*rehearsal interaction: First, according to the description, participants were tested on all learned stimuli at 12h. Why did the authors choose to do so? This is an unnecessary complication, because this means that at 24h the items in the ‘restudy’ condition actually underwent restudy and retrieval practice. It should be taken into account that under these circumstances especially the emerging differences between restudy and retrieval practice conditions at 24h is hard to interpret. Second, interpreting a difference between the two time points (“time-critical relationship between memory rehearsal and sleep-associated memory gains”) should be secured with an appropriate statistical test including both time points.

According to the methods section, participants were asked to recall the corresponding color association only if they judged a noun as old, regardless of whether it actually was an old or new item. Therefore the employed measure of memory performance is not only dependent on the ability to recall the color association in relation to the baseline test, but also on the ability to correctly recognize the item. Are there differences between the different groups/conditions in item memory that could have led to the observed differences in ‘source’ memory performance? Please include into the manuscript.

From my point of view, the conclusion that sleep preferentially strengthens implausible memories is a bit misleading. It is a pity that there is no data on performance right after the rehearsal manipulation to get a better look at the isolated effect of sleep on schema-congruency, however when comparing baseline and 12h performance in the no rehearsal condition, it seems more that sleep stabilizes plausible and implausible items alike (same performance levels) and thereby counteracts stronger forgetting of implausible information seen in the wake group.

Why do the authors call a cued recall of associated color information ‘source memory’?

How many items were judged as plausible in total and how were they distributed to the different conditions? Please report.

Reviewer #2: This is an interesting experiment that both replicates and seeks to extend prior work. It tackles timely questions and provides to the greater landscape of the sleep and memory research. The design of this study is clever and the manuscript is well written. However, there were some points that require clarification.

1. How many exposures were used in the retrieval and restudy sessions – is it just 1 additional restudy of all images/1 additional retrieval test? What did subjects do during the no-rehearsal control session? Please provide greater detail about these methods. Likewise, were participants asked if they actively tried to rehearse at all during the 12 hour intervals?

2. Is the feedback on the baseline test only an overall score or are participants informed of which trials they got wrong? It seems possible that this feedback session could contribute to the restudy advantage, providing increased motivation to re-encode the pairs they have forgotten during the restudy. Could this potentially explain why immediately there is a restudy > retrieval effect, despite no interaction with sleep group?

3. How does source memory performance change after the retrieval rehearsal session? If the interpretation of the restudy vs retrieval benefit is that retrieval facilitates the rapid consolidation of information compared to restudy, it might provide nice additional support to test if there was a gain from baseline to post-retrieval, and perhaps not between post-retrieval to the 12 hour delay test.

4. For the 12 hour Group * Plausibility interaction: It would be informative to test if there is a significant difference between the implausible pairs, but not the plausible pairs, across the Sleep and Wake groups. If so, it seems like that comparison would provide better support for the idea that sleep, and not wake, specifically benefits the implausible pairs, whereas focusing on the reduced difference between the pair types says less about which items actually benefit.

5. Were there any effects of Rehearsal Strategy using the exploratory ANOVA with the recalculated 24 vs 12 hr retention? It seems with the emergence of the Group x Rehearsal interaction only at 24 hours, a more direct comparison across the two delays would be helpful for understanding if this is an effect that only emerges over an extended period of time.

6. On page 11, the authors state that the retention performance was higher for implausible than plausible associations in the wake group. Since this is from the exploratory data analysis using the 24/12hr retention measure, it is difficult to unpack this pattern from Figure 4, which seems to show a consistent plausible > implausible pattern at both time points and seemingly gains in both across the delays. Consider adding another Figure plotting the data from the exploratory analysis.

7. Likewise, for the exploratory analysis: the authors could provide greater discussion reconciling the non-significant Group main effect just at 24 hours with the significant Group interaction in this exploratory analysis (Wake > Sleep). Is this due to gains in the Wake group over the two sessions or forgetting in the sleep group? This seems central to the claims that the Wake group is getting a boost from the intervening sleep period.

8. Greater discussion attempting to reconcile the two sets of findings and what the results tell us more broadly about when and what sleep benefits is needed. Although it’s understandable that the interaction between schema consistency and rehearsal strategy was not statistically significant, there is still room to integrate more side by side discussion of the findings. For example, why might the benefit of sleep on restudied items be time-limited but not the plausibility effects? And similarly, why might the schema effects not be susceptible to the same potential ceiling effects at the 12 hour delay?

9. The authors argue that the results suggest restudy benefits sleep only if the sleep period comes soon after encoding. However, these effects could also be due to time of day benefits or interference over the wake-filled delay. If there is enough variability in the reported onset of sleep time, the authors could attempt to quantify the effect, seeing if perhaps a median split of the sleep group shows greater restudy benefits if sleep onset is closer to the end of the restudy session. However, I realize this may not be feasible if there isn’t enough variability/power.

10. The authors do not report if there was a significant Rehearsal Strategy x Plausibility interaction, separate from group (page 12). I assume none of the statistics from that analysis were significant, but including mention of the other findings from that ANOVA would be beneficial.

11. Though the authors note that the order of the restudy and retrieval practice was counterbalanced across subjects, it would be interesting to know if there were any interesting carry over effects in this subject sample (e.g., restudy first leads to greater benefits vs. retrieval first).

12. Missing citations in introduction: in the first paragraph, citations are needed for the line stating: ‘in the short term restudied information is better remembered than information subjected to retrieval practice’.

13. Typo page 13, second sentence of discussion (" participants in the wake group had themselves had...")

6. PLOS authors have the option to publish the peer review history of their article (what does this mean?). If published, this will include your full peer review and any attached files.

Reviewer #1: No

Reviewer #2: No

---

## [Author Response · Author response to Decision Letter 0]

17 Mar 2022

Please see the attached response to Reviewers document.

---

## [Decision Letter · Decision Letter 1]

11 Apr 2022

PONE-D-21-35298R1Sleep bolsters schematically incongruent memoriesPLOS ONE

Dear Dr. Cairney,

Thank you for submitting your manuscript to PLOS ONE. After careful consideration, we feel that it has merit but does not fully meet PLOS ONE’s publication criteria as it currently stands. Therefore, we invite you to submit a revised version of the manuscript that addresses the points raised during the review process. The revised manuscript has been assessed by both original reviewers. Reviewer #1 still raises two substantial concerns regarding the results and interpretation, which should be addressed fully and with maximal transparency before the manuscript can be considered for publication.

We look forward to receiving your revised manuscript.

Kind regards,

Maria Wimber

Academic Editor

PLOS ONE

Journal Requirements:

Additional Editor Comments (if provided):

The revised manuscript has been assessed by both original reviewers. Reviewer #1 still raises two substantial concerns regarding the results and interpretation, which should be addressed fully and with maximal transparency before the manuscript can be considered for publication.

Reviewers' comments:

Reviewer's Responses to Questions

**Comments to the Author**

1. If the authors have adequately addressed your comments raised in a previous round of review and you feel that this manuscript is now acceptable for publication, you may indicate that here to bypass the “Comments to the Author” section, enter your conflict of interest statement in the “Confidential to Editor” section, and submit your "Accept" recommendation.

Reviewer #1: All comments have been addressed

Reviewer #2: (No Response)

2. Is the manuscript technically sound, and do the data support the conclusions?

Reviewer #1: Yes

Reviewer #2: Yes

3. Has the statistical analysis been performed appropriately and rigorously? 

Reviewer #1: Yes

Reviewer #2: Yes

4. Have the authors made all data underlying the findings in their manuscript fully available?

Reviewer #1: Yes

Reviewer #2: Yes

5. Is the manuscript presented in an intelligible fashion and written in standard English?

Reviewer #1: Yes

Reviewer #2: Yes

6. Review Comments to the Author

Reviewer #1: I want to thank the authors for this comprehensive revision. They have responded thoroughly to all my comments and I am happy to recommend this manuscript for publication.

Reviewer #2: Overall, I appreciate the work the authors have put into this revision and the authors have sufficiently responded to most of my initial comments. However, I do have a few remaining questions.

1. In response to my question on the first set of reviews (#5) the authors noted the analysis looking at retention between 12 and 24 hour memory tests (e.g., 24 hr hits as proportion of 12 hr hits) did not yield a significant rehearsal strategy main effect nor a group x strategy interaction. However, this seems to counter the authors main claim is that sleep selectively benefits restudied items, and that this benefit may only be evident on the 24 hr test once potential ceiling effects decay. If so, wouldn’t the wake group, and/or the retrieval for the sleep group, be expected to show a decrease in accuracy with forgetting over time? It is possible that the lack of significant effects between the 12 to 24 hour memory is indicative the initial sleep vs. wake period following learning is critical for then what is fairly stable memory, but it’s difficult to reconcile this with the emergence of the interaction only at 24 hours – what then is driving the effect? I think the authors should mention the results of this analysis explicitly, and address how it can be interpreted in the context of the other results.

2. For the Group x Rehearsal interaction at the 24 hour test: the authors note that the only the sleep group shows a restudy > retrieval effect. Does the sleep group (and/or the wake group) also show a benefit over the no rehearsal items, or is it specific to the restudy vs retrieval? If the wake group still shows a general benefit over the no rehearsal items, it still suggests any rehearsal conveys some benefit, but would speak further to the specific interaction of restudy and sleep. Please report the full results.

3. Item memory: The authors note the overall sleep vs wake effects for item memory, but do not analyze if there are any effects of rehearsal strategy or plausibility on item memory. It could be compelling if the authors could test if the reported effects are limited to associative source memory measures and are not seen for item memory itself. However, as this would be a full additional set of analyses, I leave it up to the authors’ discretion to include this suggestion.

7. PLOS authors have the option to publish the peer review history of their article (what does this mean?). If published, this will include your full peer review and any attached files.

Reviewer #1: **Yes: **Svenja Brodt

Reviewer #2: No

---

## [Author Response · Author response to Decision Letter 1]

8 May 2022

Please see the attached Response to Reviewers document.

---

## [Decision Letter · Decision Letter 2]

23 May 2022

Sleep bolsters schematically incongruent memories

PONE-D-21-35298R2

Dear Dr. Cairney,

We’re pleased to inform you that your manuscript has been judged scientifically suitable for publication and will be formally accepted for publication once it meets all outstanding technical requirements.

Kind regards,

Maria Wimber

Academic Editor

PLOS ONE

Additional Editor Comments (optional):

Reviewers' comments:

Reviewer's Responses to Questions

**Comments to the Author**

1. If the authors have adequately addressed your comments raised in a previous round of review and you feel that this manuscript is now acceptable for publication, you may indicate that here to bypass the “Comments to the Author” section, enter your conflict of interest statement in the “Confidential to Editor” section, and submit your "Accept" recommendation.

Reviewer #2: All comments have been addressed

2. Is the manuscript technically sound, and do the data support the conclusions?

Reviewer #2: Yes

3. Has the statistical analysis been performed appropriately and rigorously? 

Reviewer #2: Yes

4. Have the authors made all data underlying the findings in their manuscript fully available?

Reviewer #2: Yes

5. Is the manuscript presented in an intelligible fashion and written in standard English?

Reviewer #2: Yes

6. Review Comments to the Author

Reviewer #2: (No Response)

7. PLOS authors have the option to publish the peer review history of their article (what does this mean?). If published, this will include your full peer review and any attached files.

Reviewer #2: No

---

## [Editor Report · Acceptance letter]

15 Jun 2022

PONE-D-21-35298R2 

Sleep bolsters schematically incongruent memories 

Dear Dr. Cairney:

I'm pleased to inform you that your manuscript has been deemed suitable for publication in PLOS ONE. Congratulations! Your manuscript is now with our production department. 

Kind regards, 

on behalf of

Prof. Maria Wimber 

Academic Editor

PLOS ONE